# Gut Microbiota and Derived Short-Chain Fatty Acids Are Linked to Evolution of Heart Failure Patients

**DOI:** 10.3390/ijms241813892

**Published:** 2023-09-09

**Authors:** Javier Modrego, Adriana Ortega-Hernández, Josebe Goirigolzarri, María Alejandra Restrepo-Córdoba, Christine Bäuerl, Erika Cortés-Macías, Silvia Sánchez-González, Alberto Esteban-Fernández, Julián Pérez-Villacastín, María Carmen Collado, Dulcenombre Gómez-Garre

**Affiliations:** 1Laboratorio de Riesgo Cardiovascular y Microbiota, Hospital Clínico San Carlos, Instituto de Investigación Sanitaria San Carlos (IdISSC), 28040 Madrid, Spain; javier.modrego@salud.madrid.org (J.M.); adriana.ortega@salud.madrid.org (A.O.-H.); ssgonzale@salud.madrid.org (S.S.-G.); 2Centro de Investigación Biomédica en Red de Enfermedades Cardiovasculares (CIBERCV), Instituto de Salud Carlos III, 28029 Madrid, Spain; jvillacastin@secardiologia.es; 3Servicio de Cardiología, Hospital Clínico de San Carlos, Instituto de Investigación Sanitaria San Carlos (IdISSC), 28040 Madrid, Spain; josebe.goirigolzarri@salud.madrid.org (J.G.); alejandra.restrepo@salud.madrid.org (M.A.R.-C.); 4Instituto de Agroquímica y Tecnología de los Alimentos (IATA-CSIC), 46980 Paterna, Spain; christine.bauerl@iata.csic.es (C.B.); ercorma@iata.csic.es (E.C.-M.); mcolam@iata.csic.es (M.C.C.); 5Servicio de Cardiología, Hospital Universitario Severo Ochoa, 28914 Leganés, Spain; athalbertus@gmail.com; 6Departamento de Medicina, Facultad de Medicina, Universidad Complutense, 28040 Madrid, Spain; 7Fundación para la Investigación Interhospitalaria Cardiovascular, 28008 Madrid, Spain; 8Departamento de Fisiología, Facultad de Medicina, Universidad Complutense, 28040 Madrid, Spain

**Keywords:** heart failure, gut microbiota, short chain fatty acids, inflammation, endothelial dysfunction

## Abstract

There is a lack of direct evidence regarding gut microbiota dysbiosis and changes in short-chain fatty acids (SCFAs) in heart failure (HF) patients. We sought to assess any association between gut microbiota composition, SCFA production, clinical parameters, and the inflammatory profile in a cohort of newly diagnosed HF patients. In this longitudinal prospective study, we enrolled eighteen newly diagnosed HF patients. At admission and after 12 months, blood samples were collected for the assessment of proinflammatory cytokines, monocyte populations, and endothelial dysfunction, and stool samples were collected for analysis of gut microbiota composition and quantification of SCFAs. Twelve months after the initial HF episode, patients demonstrated improved clinical parameters and reduced inflammatory state and endothelial dysfunction. This favorable evolution was associated with a reversal of microbiota dysbiosis, consisting of the increment of health-related bacteria, such as genus *Bifidobacterium*, and levels of SCFAs, mainly butyrate. Furthermore, there was a decrease in the abundance of pathogenic bacteria. In vitro, fecal samples collected after 12 months of follow-up exhibited lower inflammation than samples collected at admission. In conclusion, the favorable progression of HF patients after the initial episode was linked to the reversal of gut microbiota dysbiosis and increased SCFA production, particularly butyrate. Whether restoring butyrate levels or promoting the growth of butyrate-producing bacteria could serve as a complementary treatment for these patients deserves further studies.

## 1. Introduction

Heart failure (HF) is a global public health problem that affects over 64 million people worldwide, and its prevalence is increasing, making it the leading cause of hospitalizations among adults over 65 years of age [1]. Many efforts made in the search for new and more effective strategies for preventing and modifying the course of HF are still not entirely satisfactory [2], since the pathophysiology of HF remains incompletely understood. Consequently, there will be an increase in the total number of hospitalizations, readmissions, and outpatient visits in the coming years.

Several studies have identified a relationship between the presence of low- and medium-grade systemic inflammation and subsequent endothelial dysfunction as pathological mechanisms associated with HF [3,4]. HF patients present a permanent elevation of circulating inflammatory cytokines (such as TNF-α, IL-1β, IL-6, and adhesion molecules), which have been related to worsening in cardiac structure and function [3]. In addition, HF patients often exhibit elevated circulating levels of monocytes, which can differentiate into pro-fibrotic macrophages within the heart [3]. Furthermore, they may experience endothelial dysfunction, which can influence cardiomyocyte function by promoting cardiac fibrosis and inflammation [4], thereby contributing to a vicious cycle. Thus, endothelial dysfunction has been reported to be an independent predictor of hospitalization and mortality in HF patients [4].

New evidence shows that human gut microbiota are essential for maintaining host health [5]. A significant loss of gut microbiota biodiversity (dysbiosis) has been reported in Western countries and has been associated with numerous pathological disorders, including cardiovascular diseases [6]. Regarding HF, several studies have found significant differences in diversity and bacterial taxa richness in both decompensated and stable HF patients, with or without reduced ejection fraction, compared to healthy individuals [7,8]. More than 90% of the gut microbiota in healthy individuals belong to two phyla (Firmicutes and Bacteroidota), followed by Proteobacteria, Actinobacteriota and Verrucomicrobiota [9]. The gut microbiota composition of HF patients is characterized by two major changes: an increment in bacteria belonging to the phylum Proteobacteria (mainly pathogenic bacteria) and a decrease in beneficial bacteria [10,11]. In this sense, several studies have demonstrated that HF patients show a higher abundance of pathogenic bacteria, including but not limited to *Salmonella*, *Shigella*, *Escherichia*, *Campylobacter*, *Klebsiella*, *Yersinia*, and *Clostridium* [11,12,13]. Some of these bacteria have been associated with the severity of the disease as expressed by New York Heart Association (NYHA) functional classes, and even have been proposed as potential prognostic markers [12,14]. In addition, *Clostridium difficile* infection in patients with HF was associated with significantly higher hospital mortality rates [13]. Furthermore, in comparison with healthy subjects, HF patients are also characterized by less abundance of short-chain fatty acid (SCFA)-producing bacteria such as *Blautia*, *Erysipelotrichaceae*, *Collinsella*, *Ruminococcaceae*, *Lachnospiraceae*, *Faecalibacterium*, *Eubacterium rectale*, and *Dorea Longicatena* [10,15,16,17,18].

Gut dysbiosis observed in HF patients has been associated with increased intestinal permeability, which eventually contributes to increased inflammatory states [19]. Gut microbiota function like an endocrine organ by generating bacterial metabolites, and gut dysbiosis may lead to changes in their production. SCFAs, mainly acetate, propionate, and butyrate, are produced by gut microbiota from fermentation of dietary fibers and resistant starch in the colon and have gut-protective effects [20]. SCFAs, mainly butyrate, are an energy source for colonocytes, and exert an anti-inflammatory effect by upregulating tight junctions in the intestinal cells that reduce intestinal permeability [21]. In addition, it has been shown that SCFAs enhance the proliferation and activation of immune cells [22]. Several studies have observed a decrease in gut microbiota richness and butyrate-producing bacteria in cardiovascular diseases [23]. HF patients show decreased levels of SCFA-producing bacteria [10,15,16,17,18], suggesting a low capacity to produce SCFAs in these patients. Although gut microbiota composition has been reported to be different in HF patients with preserved left ventricular ejection fraction (pLVEF) and with reduced left ventricular ejection fraction (rLVEF) [16,17], the loss of SCFA-producing bacteria is similar regardless of the type of HF [10]. Additionally, Cui et al. have reported an imbalance of gut microbes involved in the production of protective metabolites such as butyrate and harmful metabolites such as trimethylamine N-oxide (TMAO) [17].

Despite this information, direct evidence of the association between gut microbiota dysbiosis and changes in bacterial metabolites in HF patients is lacking, and follow-up studies are needed to investigate the causal relationship. Our group has demonstrated an association between specific changes in gut microbiota and the development of HF in a hypertensive model of HF [24]. Therefore, in this study, we have investigated gut microbiota composition and its association with clinical parameters, the production of SCFAs, and the inflammatory profile of a cohort of de novo HF patients followed during the 12 months after admission.

## 2. Results

### 2.1. Clinical, Inflammatory, and Endothelial Characteristics of Patients

Eighteen patients with newly diagnosed HF were included in the study and followed up for 12 months. The mean age was 67.6 years, and 61.1% were women. Their clinical characteristics at admission and at 12-month follow-up are shown in Table 1. Of the 18 patients, 44% were current smokers and 38.9% had T2DM; dyslipidemia was present in 55.6% of patients. A small percentage of patients (11.1%) already showed coronary heart disease (CHD). On admission, NT-proBNP was 7,081 pg/mL, and only seven patients (38.9%) had HF with pLVEF (LVEF > 40%). Two patients died during the emergence of the COVID-19 pandemic before the 12-month sample could be collected. The remaining patients were asymptomatic at the 12-month follow-up, with normal NT-proBNP levels (358 ± 69 pg/mL) and significant recovery in LVEF (more than 81% of patients showed pLVEF). Over 93% of patients demonstrated NYHA class I/II symptoms. These data are presented in Table 1.

It is well known that HF patients show a low-medium systemic inflammatory grade and subsequent endothelial dysfunction that correlate with HF severity [4]. In our study, the circulating levels of proinflammatory biomarkers, such as ICAM-1, IL-6, IL-18, c-reactive protein (CRP), sCD163, TNF-α and VCAM-1, decreased significantly at the 12-month follow-up compared to the time of admission (Table 2). Circulating intermediate (CD45+/CD14+/CD16+) and non-classical (CD45+/CD14^low^/CD16+) monocyte populations, considered activated monocytes, decreased after 12 months, although they did not reach statistical significance (Table 2). However, classical non-activated monocytes (CD45+/CD14+/CD16^−^) increased significantly over time (Table 2). By contrast, the levels of circulating endothelial progenitor cells (EPCs) (CD45+/CD34+/KDR+ and CD45+/CD34+/VE-cadherin+), which have been shown to serve as a cellular reservoir for replacing dysfunctional endothelium [25], significantly increased over time (Table 2), suggesting a regeneration of endothelial cells.

### 2.2. Gut Microbiota Composition Associated with the Course of HF

To analyze gut microbiota composition, we first calculated α diversity, a parameter that takes into account the intrinsic biodiversity of each sample, and β diversity, which measures the (dis)-similarity between samples. For this purpose, we rarefied the sequence depth to 55,701 read counts in all samples. No significant differences were found in either α diversity (Simpson, Shannon, and Pielou indexes) (Figure 1a–c) or β diversity (Figure 1d) between admission and the 12-month follow-up.

We further investigated whether there were any changes in the taxonomic composition of the gut microbiota. Both at admission and after 12 months, the most abundant phyla were Bacteroidota, Firmicutes, and Proteobacteria, accounting for around 97% of bacterial taxa. Only the abundance of phylum Proteobacteria showed a significant change after 12 months of follow-up, with a decrease compared to the level at the time of admission (Figure 2a). Among the less abundant phyla, there was an increase in phyla Actinobacteriota and Verrucomicrobiota, while phylum Fusobacteriota decreased, although without statistical significance (Figure 2b).

We also investigated whether the course of HF was associated with more specific changes in the gut microbiota composition. The top thirty bacteria genera differentially expressed between admission and after 12 months of follow-up are displayed in Figure 3a. We identified seven taxa with significantly different abundance. In particular, *Acidaminococcus* (phylum Firmicutes) and *Bifidobacterium* (phylum Actinobacteriota) genera increased after 12 months of follow-up, while *Pectobacterium*, *Sphingosinicella*, *Sphingomonas*, and *Bradyrhizobium* (all phylum Proteobacteria), and RF39 (phylum Firmicutes) decreased (Figure 3a). Other genera such as *Akkermansia* (phylum Verrucomicrobiota), *Collinsella* (phylum Actinobacteriota), *Faecalibacterium*, *Lachnospira*, *Roseburia*, *Ruminococcus*, and *Subdoligranulum* (all phylum Firmicutes), and *Odoribacter* and *Parabacteroides* (phylum Bacteroidota) increased, and some such as *Enterobacter* (phylum Proteobacteria) decreased at 12 months after admission, although without reaching statistical significance (Figure 3a). We used the LEfSe analysis to elucidate which genera were most likely accountable for disparities between the admission and 12-month groups. We found that high abundance of *Acidaminococcus* and *Bifidobacterium* were characteristic of the 12-month group, while *Bradyrhizobium*, *Spingosinicella*, and *Sphingomonas* were characteristic of the admission group (Figure 3b).

Interestingly, the abundance of some of these genera were already significantly changed six months after admission (Figure 4).

To ensure that the dietary habits of patients were not a confounding factor, compliance with the Mediterranean Diet was quantified at admission and after 12 months of follow-up. No significant changes were identified during the study period [admission score: 8 (6–9), 12-month score: 8 (7–10); *p* = NS].

### 2.3. Fecal SCFA Concentration Associated with HF Course

To investigate the ability of gut microbiota to produce SCFAs, we quantified the most abundant SCFAs in the HF patients’ stools. Acetate, propionate, and butyrate fecal concentrations were significantly higher at 12-month follow-up than at admission (Figure 5a). Both acetate and butyrate fecal concentration were already significantly higher at 6-month follow-up (Figure 5a). This increment in fecal SCFA concentrations was associated with an increase in the main SCFA-producing bacteria (Figure 5b, Table 3).

### 2.4. Impact of Taxonomic Biomarkers and SCFAs on Clinical, Inflammatory and Endothelial Function Markers

Correlation analysis was performed between genera differentially expressed between groups and SCFA concentration, and markers of clinical evolution (NT-proBNP, LVEF), monocyte activation (CD45+/CD14+/CD16^−^), inflammation (CRP, IL-18, IL-6, TNF-α, sCD163, ICAM-1 and VCAM-1), and endothelial function (EPCs CD45+/CD34+/KDR+ and CD45+/CD34+/VE-cadherin+) (Figure 6). *Acidaminococcus* and *Bifidobacterium* negatively correlated with proinflammatory biomarkers (VCAM-1 and CRP, respectively), and positively with endothelial function markers (EPC CD45+/CD34+/KDR+ and CD45+/CD34+/VE-cadherin+). *Bifidobacterium* also correlated negatively with circulating NT-proBNP levels (Figure 6). By contrast, the abundance of *Bradyrhizobium*, *Sphingomonas* and *Sphingosinicella* genera correlated positively with NT-proBNP and proinflammatory markers (IL-18, TNF-α and VCAM-1), and negatively with LVEF, anti-inflammatory monocytes (CD45+/CD14+/CD16^−^), and EPCs CD45+/CD34+/KDR+, and CD45+/CD34+/VE-cadherin+ levels (Figure 6).

Regarding SCFAs, butyrate concentration in stool samples negatively correlated with NT-proBNP plasma levels and proinflammatory parameters (ICAM-1), and positively with anti-inflammatory monocytes (CD45+/CD14+/CD16^−^) and EPCs (CD45+/CD34+/KDR+ and CD45+/CD34+/VE-cadherin+). A negative correlation was also observed between acetate and NT-proBNP levels, and a positive correlation was observed between LVEF and EPCs CD45+/CD34+/VE-cadherin+. Propionate correlated positively with anti-inflammatory monocytes (CD45+/CD14+/CD16) (Figure 6).

### 2.5. Effect of HF Patients’ Fecal Supernatant on Intestinal Inflammation

We then characterized the direct impact of fecal supernatants from HF patients on intestinal inflammation and cell viability using colonic HT29 NF-κB reporter cells. The NF-κB inflammatory response was higher after 24 h exposure of colonic HT29 cells to fecal samples collected at admission than with those collected after 12 months of follow-up (Figure 7a). This decrease in the inflammatory response was already evidenced after 6 months of follow-up (Figure 7a). The viability of colonic cells HT-29 incubated with fecal samples of HF patients did not change significantly throughout the follow-up (Adm: 96.2 ± 3.4%; 6 m: 100.1 ± 2.5%; 12 m: 104.2 ± 3.7%). Interestingly, genus *Bifidobacterium* abundance correlated negatively with NF-κB-mediated HT29 inflammation (Figure 7b).

## 3. Discussion

To our knowledge, this is the first prospective longitudinal study investigating the association between gut microbiota dysbiosis and changes in bacterial metabolites in HF patients. Our data show that improvement of clinical parameters and the inflammatory state 12 months after a first episode of HF is associated with the restoration of the gut microbiota composition, characterized by an increment in the abundance of health-related bacteria and SCFAs, and a diminution of the presence of proteolytic bacteria. Accordingly, in vitro, fecal samples collected after 12 months of follow-up induced lower NF-κB activation than those collected at admission, corroborating the loss of their proinflammatory capacity.

Healthy gut microbiota is mainly composed of bacteria belonging to the phyla Firmicutes, Bacteroidota, and, in a lesser proportion, to the phyla Proteobacteria, Actinobacteriota, and Verrucomicrobiota [6], and changes in this microbial composition have been demonstrated to cause and/or worsen several pathological conditions, including cardiovascular diseases such as HF [7]. In this sense, previous studies have shown that gut microbiota of HF patients, in particular those in a decompensated phase, are characterized by an increment in phylum Proteobacteria (mainly pathogenetic bacteria), and a decrease in bacteria with anti-inflammatory functions [12]. Interestingly, the increase in pathogenic bacteria such as *Campylobacter*, *Shigella*, *Salmonella*, and *Yersinia* has been associated with low-grade inflammation in chronic HF and correlated with worsened NYHA class [7,12]. Previous studies have also demonstrated a depletion of genera with a protective role, such as *Akkermansia*, *Faecalibacterium*, or *Blautia*, in the gut microbiota of HF patients. *Akkermansia* is the most important modulator of mucus secretion and gut epithelial integrity, *Faecalibacterium* is a key butyrate-producing genus, and *Blautia* has been associated with anti-inflammatory mechanisms, having been proposed as a potential probiotic [37]. Consistent with previous data, in this study, HF patients showed higher abundance of bacteria of phylum Proteobacteria and lower abundance of bacteria of phylum Actinobacteriota at admission than after 12 months of follow-up. Specifically, the genera *Bradyzhizobium*, *Spingomonas* and *Spingosinicella* (belonging to phylum Proteobacteria) were identified as characteristic taxa of HF patients at admission. Several studies have found an increment in *Bradyrhizobium* in patients with cardiovascular disease, including individuals with aortic aneurysm [38,39]. Jing et al. have reported the presence of *Spingomonas* in the blood microbiome of hypertensive patients [40]. Phylum Proteobacteria is composed of gram-negative bacteria, and its outer membrane contains lipopolysaccharide (LPS), one of the most effective inflammatory inducers. Studies show that LPS is significantly associated with HF aggravation [41]. In this sense, it has been suggested that the interaction between gut microbiota and the immune system may contribute to systemic inflammation in HF [42]. HF patients show reduced cardiac output leading to worsening intestinal perfusion, and in consequence, to increased gut permeability, that can favor LPS translocation into systemic circulation [43]. This low-grade endotoxaemia has effects on several cell types, such as monocytes or endothelial cells, shifting them to a proinflammatory state that contributes to HF [44]. In fact, proinflammatory cytokine such as TNF-α, IL-1β, IL-6, adhesion molecules, and immune cells induce fibrosis and influence cardiac function, and are correlated with worse prognoses [45]. In our study, the favorable clinical evolution of HF patients was associated with a depletion of proinflammatory bacteria and with an increment of bacteria genera with a protective role, such as *Akkermansia*, *Faecalibacterium*, *Blautia*, or *Bifidobacterium*, which were the characteristic taxa of the 12-month group. Genus *Bifidobacterium* has been demonstrated to improve intestinal barrier function and modulate the secretion of proinflammatory cytokines (such as TNF-α), which may improve cardiovascular diseases [46,47]. Further in vitro studies will elucidate the functional role of these genera in HF evolution.

A dysbiotic gut microbiota has been linked to a reduction in the production of SCFAs, which have important beneficial properties for human health [47]. Acetate, propionate, and butyrate concentrations were significantly higher in HF patients after 12 months of follow-up than at admission. Increasing levels of butyrate were associated with lower levels of NT-proBNP and inflammatory proteins and higher levels of circulating EPCs, which have been linked to a lower risk of developing cardiovascular disease [25]. SCFAs, particularly butyrate, have been related to inflammation reduction, epithelial barrier integrity maintenance, stimulation of immune cells and protection against cardiovascular disease [20]. Butyrate is an important regulator of intestinal barrier function by increasing the expression of tight junction proteins such as ZO-1, claudin-1, and occludin [48]. In addition, butyrate is a potent inhibitor of the transcription factor NF-κΒ, one of the most important regulators of proinflammatory gene expression, permanent activation of which may favor HF progression [49]. It has been demonstrated that butyrate can downregulate the production of LPS-induced proinflammatory cytokines in immune cells such as monocytes and macrophages [20]. It has also been reported that propionate and butyrate may induce the differentiation of regulatory T cells, which inhibit both inflammation and HF progression, through epigenetic mechanisms [50].

In our study, the loss of the ability of the gut microbiota of HF patients to induce an inflammatory response was confirmed by in vitro studies, since NF-κB activation was higher in colonic HT29 cells exposed to fecal samples collected at admission than in those collected after 12 months of follow-up. The proinflammatory ability of fecal samples from HF patients negatively correlated with the abundance of genus *Bifidobacterium*, confirming previous studies that have demonstrated that *Bifidobacterium* can inhibit the NF-κB pathway [51,52]. Unfortunately, in this study we have not investigated which components of the fecal supernatant contribute to the inflammatory response. Thus, the establishment of a mechanistic link between gut microbiota and HF requires further investigation.

In recent years, several studies have investigated the mechanisms of butyrate at the cellular and systemic levels and its potential therapeutic use in different pathologies [53,54]. Whether HF patients may benefit from the adjuvant treatment with butyrate will be confirmed with large designed clinical trials. From a clinical perspective, new research indicates that SCFAs may have a significant impact on the development of HF, possibly due to their anti-inflammatory properties. However, the extent to which they can be utilized in treating HF patients remains unclear. Our findings emphasize the connection between the improvement in HF patients’ condition following their initial episodes and the reversal of gut microbiota dysbiosis and the restoration of levels of butyrate. As a result, our study lays the groundwork for future human clinical trials seeking to evaluate the supplementary use of butyrate as a treatment for HF patients.

Our study has some limitations. First, the population size is relatively small, although it closely resembles that of previous studies [8,11,12,15,17,18]. Nonetheless, we believe that the longitudinal study design, combined with cell culture experiments, enhances the robustness of our findings. Second, all patients had a favorable recovery after 12 months, without hospitalization or deaths. Since we were unable to determine the cause of death for the two patients who died during the emergence of the COVID-19 pandemic, they were excluded from the analysis. Therefore, gut microbiota changes in patients with unstable HF disease deserve further study.

In conclusion, our results indicate that patients with a first diagnosis of HF and favorable evolution during the first year of follow-up show significant reversal of gut microbiota dysbiosis documented at the time of admission. Moreover, clinical improvement of HF was associated with an increase in the production of SCFAs, mainly butyrate, and a decrease in the proinflammatory state. Our results warrant further investigation of the potential benefit of coadjuvant therapies aiming to modulate gut microbiota and its anti-inflammatory metabolites in HF patients.

## 4. Materials and Methods

### 4.1. Study Population

This was a prospective study including consecutive patients older than 18 years admitted to the Cardiology Service of the Hospital Clínico San Carlos between November 2018 and January 2021 with a first diagnosis of decompensated HF. All patients were assessed within 24 h of admission by study investigators to confirm the diagnosis of HF based on the 2016 ESC Guidelines for the diagnosis and treatment of acute and chronic HF [55], and to confirm that it was the first episode of HF. We excluded patients with inflammatory/autoimmune diseases or cancer or who had been treated with antibiotics and/or probiotics within the previous two months.

We collected demographic and clinical data and past medical history from all patients. In addition, a transthoracic echocardiogram, a chest radiograph, and a determination of NT-ProBNP were performed. All patients were hospitalized for at least 24 h, and during this time they were managed according to the clinical guidelines of the Cardiology Service. In addition, participants completed the 14-item questionnaire of the PREDIMED study at admission and after 12 months of follow-up to assess their adherence to the Mediterranean diet [56]. Scores ranged from low (0–5), medium (6–9) and high (10–14), with higher scores indicating greater adherence to the Mediterranean diet.

This work has been carried out in compliance with the fundamental principles established in the Declaration of Helsinki and was approved by our local Research Ethics Committee (C.P. CB16/11/00276–C.I. 18/230-E). All patients included in the study signed a written statement of informed consent.

### 4.2. Plasma Collection and Quantification of Proinflammatory Biomarkers

Peripheral blood samples were obtained from each patient at admission and after 12 months by venipuncture in vacuum tubes using EDTA as an anticoagulant. Plasma was obtained by centrifuging the blood at 250× *g* for 10 min. The samples were then divided into aliquots and stored at −80 °C until use. Circulating levels of D-dimer, ICAM-1, IL-1β, IL-6, IL-18, CRP, sCD14, sCD163, TNF-α, and VCAM-1 were determined in plasma by automated ELISA (Protein SimplePlex Assay, Bio-Techne, Minneapolis, MN, USA) following the instructions from the manufacturer. The sensitivities of the assays were 36.4, 2.01, 0.064, 0.260, 0.288, 1.24, 2.78, 318, 0.278, and 53.7 pg/mL, respectively.

### 4.3. Flow Cytometry Analysis

A blood sample (100 µL) was stained with specific antibodies: anti-CD45 PE/Cy5.5^®^ (PC5)-conjugated (mouse IgG1a, Beckman Coulter, Brea, CA, USA), anti-CD16 Fluorescein-5-isothiocyanate (FITC)-conjugated (mouse IgG1, Beckman Coulter), and anti-CD14 PE/Cy7^®^ (PC7)-conjugated (mouse IgG2a, Beckman Coulter) for the identification of different monocyte subsets. For EPC analysis, blood samples were stained with anti-CD34 FITC-conjugated antibody (mouse IgG1, Beckman Coulter), anti-CD45 PC5-conjugated antibody (mouse IgG1a, Beckman Coulter), anti-KDR PC7-conjugated antibody (mouse IgG1, BioLegend, San Diego, CA, USA), and anti-VE-cadherin (CD144) Phycoerythrin (PE)-conjugated antibody (mouse IgG2b, R&D Systems, Minneapolis, MN, USA). Appropriate unstained samples were used for negative control assay. Incubations were performed for 30 min at 4 °C in the dark, and a lyse-wash procedure was then performed. Cells from both experiments were acquired on a Cytoflex flow cytometer (Beckman Coulter). For each sample, a minimum of 100,000 CD45+ events were acquired. Different monocyte subsets were identified as CD45+/CD14+/CD16^−^ (classical), CD45+/CD14+/CD16+ (intermediate), and CD45+/CD14low/CD16+ (non-classical). EPCs were identified as CD45+/CD34+/KDR+/or CD45+/CD34+/VE-cadherin+.

### 4.4. Fecal Sample Collection and DNA Extraction

Stool samples were obtained from each patient at admission and after 6 and 12 months using a tube with a DNA stabilizer (OMNIgene-GUT, DNAgenotek, Ottawa, Canada), and were stored at −80 °C until further processing. Bacterial DNA was extracted from the samples using the QIAamp Fast DNAStool mini kit (Qiagen, Hilden, Germany) following the manufacturer’s protocol. DNA integrity was assessed with a BioAnalyzer 2100 (Agilent, Palo Alto, CA, USA), and its concentration was determined with a Qubit 3.0 fluorometer using the dsDNA HS assay (Thermo Fisher Scientific, Waltham, MA, USA).

### 4.5. 16S rRNA Gene Sequencing and Bioinformatics Analysis

For each stool sample, the 16S rRNA gene was amplified by PCR using the Ion 16S Metagenomics kit (Thermo Fisher Scientific). After preparing the libraries using the Ion Plus Fragment Library Kit (Thermo Fisher Scientific) and performing molecular identification using the Ion Express Barcode Adapters Kit (Thermo Fisher Scientific), they were sequenced on an Ion S5 System (Thermo Fisher Scientific). Obtained sequences were processed and final sequences were clustered into operational taxonomic units (OTUs) with 99% identity using the SILVA 16S database (v138). To assess bacterial richness, diversity and evenness (α diversity), Simpson, Shannon and Pielou’s indices, respectively, were calculated from a rarefied OTU profile. β diversity was analyzed using the Bray–Curtis index, and principal coordinate analysis (PCoA) was applied to the resulting dissimilarity matrix. Two-dimensional PCoA plots were used to visualize the compositional structure of groups and samples. Taxonomic assignment of the referenced OTUs was carried out using QIIME 2 and the SILVA database (v138). Taxonomic relative abundance was calculated as a percentage based on the total number of reads per sample, and taxa with relative abundance < 0.001% were excluded for the analysis. Linear discriminant analysis effect size (LEfSe), an algorithm for discerning high-dimensional biomarkers and providing explanations, was used to determine the taxa most likely accountable for disparities between groups [57]. This was achieved by combining standard statistical significance tests with supplementary tests that encapsulate biological consistency and the importance of effects. In this study, differentially abundant taxa were selected between groups with an LDA score ≥ 2.0 and *p* < 0.05 [58].

### 4.6. Measurement of SCFAs

SCFA analysis was performed using gas chromatography–mass spectrometry (GC-MS), following the method described by Eberhart et al. [59]. For each fecal sample, an aliquot of 100 mg was placed in a 15 mL vial. Then, 1000 μL of the internal standard solution (3-Methylvaleric acid) was added, followed by 2 mL of diethyl ether. The vial was capped, shaken vigorously by hand and vortexed for 10 s. Subsequently, approximately 5–10 g of sodium sulfate was added to the vial. The vial was tightly recapped and shaken vigorously by hand and then vortexed for another 10 s. The samples were then centrifuged at 4000 rpm for 2 min at 4 °C. Finally, the supernatant was injected. The analysis was performed using an Agilent GC 7890B–5977B GC-MS with a multipurpose sampler (Gerstel MPS, Mülheim, Germany). The GC column used was Agilent DB-FATWAX, 30 m × 0.25 mm × 0.25 μm, operated in split mode (20:1). The oven temperature program was set as follows: 100 °C for 3 min, ramped to 100 °C at a rate of 5 °C min^−1^, then to 150 °C for 1 min, further ramped to 200 °C at a rate of 20 °C min^−1^, and finally held at 200 °C for 5 min. Helium was used as the carrier gas at a flow rate of 1 mL min^−1^, with an inlet temperature of 250 °C. The injection volume was two μL. Standards curves for acetate, butyrate, and propionate were used for quantifying the SCFAs in fecal samples.

### 4.7. In Vitro NF-kB Activation Assay

The proinflammatory effects of available HF patients’ fecal supernatants (admission = 11, 6 months = 14, 12 months = 9) were determined using the colonic HT29 cell line stably transfected with the secreted alkaline phosphatase (SEAP) reporter gene under the transcriptional control of a NF-κB response element as previously described [60]. Fecal supernatants were obtained by resuspending 100 mg of feces with 1 mL PBS, vortexed, and centrifuged at 16,000× *g* for 10 min at 4 °C. Then the resulting supernatant was filter-sterilized using a 0.20 μm filter. HT29 NF-κB reporter cells were grown in DMEM medium supplemented with 10% fetal bovine serum (FBS), 1% sodium pyruvate, 1% penicillin/streptomycin, and 200 g/mL zeocin at 37 °C and 5% CO_2_. The HT29 cells were then subcultured at 6.5 × 10^4^ cells/well in 96-well plates and incubated at 37 °C and 5% CO_2_ for 24 h. At this point, fecal supernatants in a 1:10 dilution were added, and after a 24-h incubation the cell supernatants were collected and SEAP activity measured using p-nitrophenyl phosphate following the manufacturer’s instructions (Thermo Fisher Scientific). In addition, cell viability was measured by adding 150 µL of resazurin solution (10 µg/mL) to each well and incubating at 37 °C and 5% CO_2_ for 5 h. Viable cells reduced resazurin to fluorescent resorufin, which was measured at 540–15 nm excitation and 590–20 nm emission using a CLARIOstar (BMG Labtech, Ortenberg, Germany) plate reader.

### 4.8. Statistical Analysis

The variables collected in the study, as well as the alpha diversity and abundance of the different taxa, are expressed as median (interquartile range) or mean ± standard error mean (SEM). The Shapiro–Wilk test was used to test the normality of each variable. Wilcoxon’s test was used to determine statistical significances. The differences between groups regarding the structure of the bacterial communities (β diversity) were analyzed with a permutation analysis and multiple ANOVA (PERMANOVA) with 999 permutations. Heatmaps of Spearman’s correlations and linear model regression analysis were performed in R Studio v2021.09.0 in R v4.1.2 from stats package v4.1.2. Statistical analysis was carried out using the Python SciPy package and SPSS version 21.0. For all analyses, differences were considered statistically significant when *p* < 0.05.

## 5. Conclusions

Our findings allow us to emphasize the following key points. (1) Positive clinical advancements observed within one year in new-onset HF patients are associated with the restoration of imbalanced gut microbiota; (2) certain genera of gut microbiota may function as indicators of patient progression; (3) the increased production of SCFAs, notably butyrate, is linked to improvements in clinical outcomes, inflammation levels, and endothelial function in HF patients; and (4) in vitro experiments conducted on colonic HT29 cells exposed to fecal samples validate the reduced proinflammatory capacity of the gut microbiota in HF patients after one year of follow-up.

## Figures and Tables

**Figure 1 ijms-24-13892-f001:**
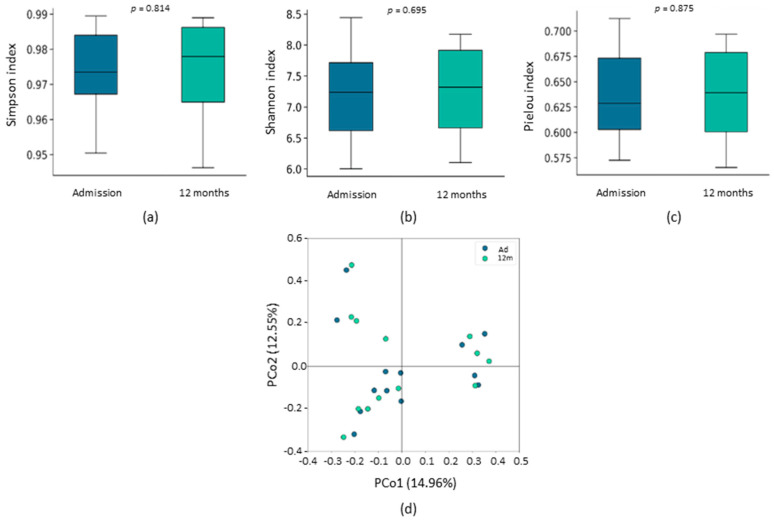
Gut microbiota diversity of HF patients at admission and after 12 months of follow-up. α diversity is presented by Simpson richness index (**a**), Shannon diversity index (**b**) and Pielou evenness index (**c**). The results are shown in box plots. β diversity (**d**) is represented by a PCoA plot of Bray–Curtis dissimilarity. PCo1 and PCo2 values for each sample are plotted with the percentage of explained variance shown in parentheses.

**Figure 2 ijms-24-13892-f002:**
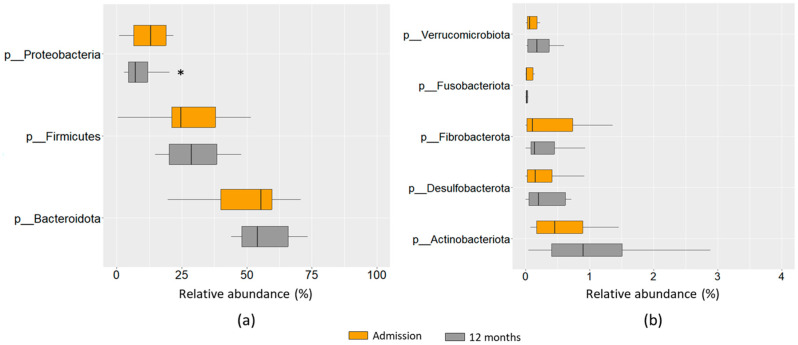
Gut microbiota composition at phylum level of HF patients. Relative abundance of majority (**a**) and minority (**b**) bacterial phyla of HF patients at admission and after 12 months of follow-up. The results are shown in box plots. *n* = 16–18, * *p* < 0.05 vs. admission.

**Figure 3 ijms-24-13892-f003:**
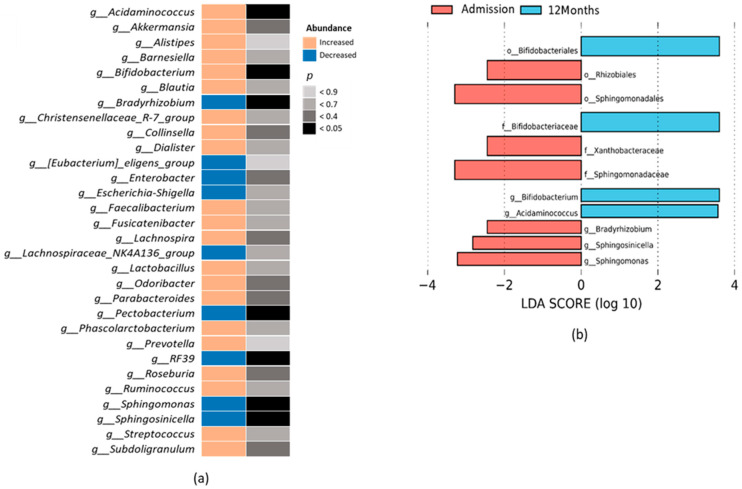
Gut microbiota composition of HF patients after 12-month follow-up. (**a**) Top 30 bacteria genera differentially expressed. Left column displays the direction of the abundance change 12 months after admission. Right column shows the level of significance. *n* = 16–18. (**b**) LEfSe analysis showing orders, families and genera differentially expressed. The length of the horizontal bars represents the LDA score (threshold log LDA score ≥ 2.0). *n* = 16–18. Abbreviations: o, order; f, family; g, genus.

**Figure 4 ijms-24-13892-f004:**
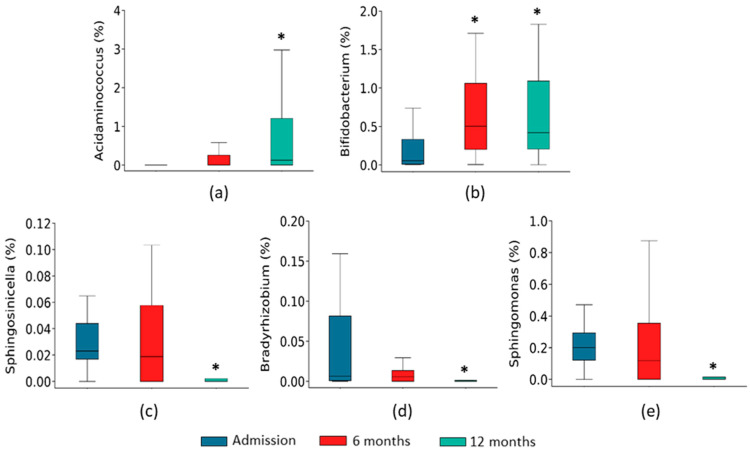
Changes of genera differentially expressed identified by LefSe analysis. Relative abundance of *Acidaminococcus* (**a**), *Bifidobacteirum* (**b**), *Sphingosinicella* (**c**), *Bradyrhizobium* (**d**), and *Sphingomonas* (**e**) in gut microbiota of HF patients at admission and after 6 and 12 months of follow-up. The results are shown in boxplots. *n* = 16–18, * *p* < 0.05 vs. admission.

**Figure 5 ijms-24-13892-f005:**
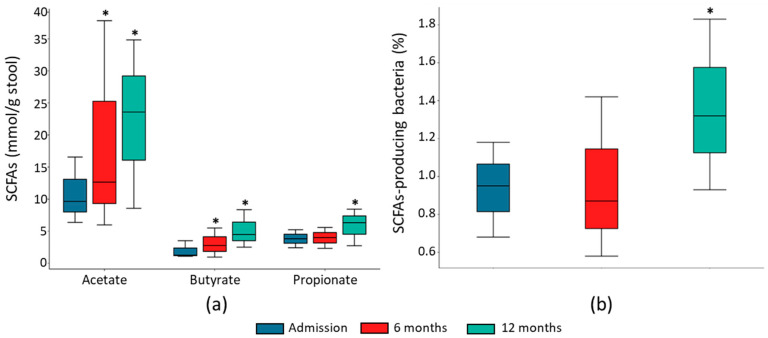
Changes in SCFAs and SCFA-producing bacteria. Fecal concentration of acetate, butyrate, and propionate (**a**), and relative abundance of SCFA-producing bacteria (**b**) in HF patients at admission and after 6 and 12 months of follow-up. The results are shown in box plots. *n* = 16–18, * *p* < 0.05 vs. admission.

**Figure 6 ijms-24-13892-f006:**
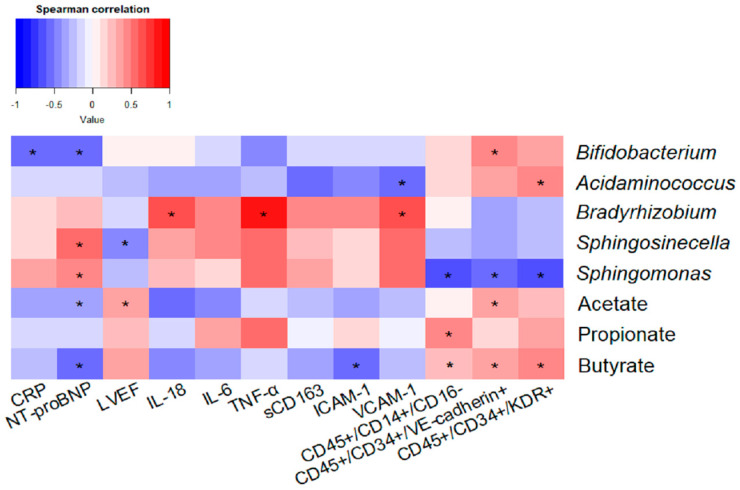
Association of gut microbiota and metabolites with clinical, inflammatory and endothelial function markers. Heatmap showing Spearman correlations of genera differentially expressed and acetate, propionate and butyrate with markers of clinical evolution, monocyte activation, inflammation and endothelial function. *n* = 16–18, * *p* < 0.05.

**Figure 7 ijms-24-13892-f007:**
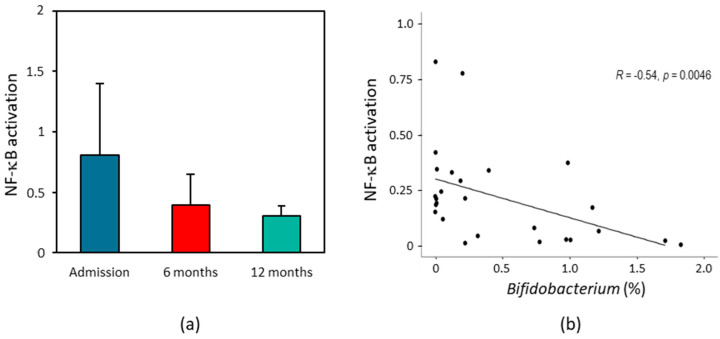
Effect of HF patients’ fecal supernatants on intestinal inflammation. (**a**) Effect of fecal samples collected at admission and after 12 months on NF-κB activation in colonic HT29 cells. The results are mean ± SEM. *n* = 7–14. (**b**) Linear model regression between NF-κB activation in colonic HT29 cells and the abundance of genus *Bifidobacterium*.

**Table 1 ijms-24-13892-t001:** Clinical characteristics of de novo HF patients.

	Admission	12 Months
Age, years	67.6 ± 4.1	68.3 ± 4.3
Male/female, n	7/11	-
Smokers, n (%)	8 (44.4)	-
T2DM, n (%)	7 (38.9)	-
Dyslipidemia, n (%)	10 (55.6)	-
CHD, n (%)	2 (11.1)	-
CVD, n (%)	1 (5.6)	-
PAD, n (%)	1 (5.6)	-
SBP, mmHg	136 ± 4	129 ± 9
DBP, mmHg	83 ± 4	77 ± 4
HR, bpm	91.3 ± 5.3	61.4 ± 2.7
NT-proBNP, pg/mL	7,081 ± 3544	358 ± 69
LVEF, %	36.2 ± 4.3	56.7 ± 3.5
Patients with LVEF > 40%, n (%)	7 (38.9)	13 (81.2)
LVEF in patients with pLVEF *, %	56.4 ± 4.3	62.7 ± 5.6
LVEF in patients with rLVEF *, %	26.5 ± 2.8	52.1 ± 4.4
NYHA class, n (%)		
I/II	0 (0)	15 (93.8)
III	12 (66.7)	1 (6.2)
IV	6 (33.3)	0 (0)
Medications, n (%)		
ACEI/ARB	7 (38.9)	13 (81.3)
Statins	7 (38.9)	8 (50.0)
Diuretics	2 (11.1)	10 (62.5)
β-blockers	4 (22.2)	14 (87.5)
MRAs	0 (0)	8 (50.0)
OADs	6 (33.3)	9 (56.3)

Values are expressed as mean ± SEM for continuous variables or n (%) for categorical variables. SBP: systolic blood pressure; DBP: diastolic blood pressure; T2DM: diabetes mellitus type 2; CHD: coronary heart disease; CVD: cerebrovascular disease; PAD: peripheral artery disease; HR: Heart rate; bpm: beats per minute; pLVEF: preserved left ventricular ejection fraction (LVEF > 40%); rLVEF: reduced left ventricular ejection fraction (LVEF ≤ 40%); NYHA: New York Heart Association classification; ACEI: angiotensin-converting enzyme inhibitor; ARB: angiotensin receptor blocker; MRA: mineralocorticoid receptor antagonists, OAD: oral antidiabetic drugs. * At admission.

**Table 2 ijms-24-13892-t002:** Inflammation and endothelial dysfunction biomarkers in HF patients over time.

	Admission	After 12 Months
Circulating proinflammatory biomarkers		
D-dimer, µg/mL	795.7 (493.2–1217.7)	563.2 (362.6–703.8)
ICAM-1, µg/mL	512.3 (480.6–596.8)	427.5 (374.9–480.2) *
IL-1β, ng/mL	0.3 (0.2–0.5)	0.3 (0.2–0.4)
IL-6, ng/mL	2.9 (2.4–4.5)	2.3 (1.6–3.4) *
IL-18, ng/mL	186 (132.3–228.3)	158 (109.9–193.5) *
CRP, mg/L	7.5 (3.7–22.2)	2.9 (2.7–5.3) *
sCD14, mg/mL	1.4 (1.3–1.6)	1.4 (1.2–1.6)
sCD163, mg/mL	0.8 (0.7–1.1)	0.6 (0.5–0.8) *
TNF-α, ng/mL	8.9 (6.4–9.6)	7.3 (5.4–8.3) *
VCAM-1, mg/mL	1.1 (0.8–1.3)	0.8 (0.7–1.0) *
Monocyte subset populations		
Classical CD45^+^/CD14^+^/CD16^−^, %	65.3 (43.7–68.0)	76.5 (66.9–87.0) *
Intermediate CD45^+^/CD14^+^/CD16^+^, %	18.7 (16.1–30.4)	16.1 (7.9–23.5)
Non-classical CD45^+^/CD14^low^/CD16^+^, %	9.5 (6.8–10.6)	4.5 (3.1–12.5)
Circulating EPCs		
CD45+/CD34^+^/VE-cadherin^+^, %	0.05 (0.02–0.06)	0.25 (0.10–0.76) *
CD45^+^/CD34^+^/KDR^+^, %	0.02 (0.01–0.04)	0.10 (0.06–0.7) *

Values are expressed as median (interquartile range). * *p* < 0.05 with respect to admission.

**Table 3 ijms-24-13892-t003:** Main SCFA-producing bacteria quantified in de novo HF patients.

SCFA	SCFA-Producing Bacteria	References
Acetate	*Akkermansia*, *Bacteroides*, *Bifidobacterium*, *Lactobacillus*, *Prevotella*, *Ruminococcus*	[26,27,28,29,30,31]
Butyrate	*Anaerostipes*, *Butyricicoccus*, *Butyricimonas*, *Butyrivibrio*, *Coprococcus*, *Eubacterium*, *Faecalibacterium*, *Flavonifractor*, *Odoribacter*, *Roseburia*	[32,33,34,35,36]
Propionate	*Akkermansia*, *Roseburia*, *Veillonella*, *Phascolarctobacterium*, *Bacteroides*, *Coprococcus*, *Dialister*, *Prevotella*	[36]

## Data Availability

All data supporting the findings of this study are available within the paper.

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
