# Peer review of "Gut Microbiota and Derived Short-Chain Fatty Acids Are Linked to Evolution of Heart Failure Patients"

_ijms, 2023, doi:10.3390/ijms241813892_

Round 1
Reviewer 1 Report
Dear authors,
I appreciate the opportunity to review this article. The authors have conducted a prospective, longitudinal study to establish relationships between SCFA and dysbiosis in HF patients.
In the Introduction, the authors give a brief overview of this current topic. However, it is important to note that this field of study of growing interest is rapidly developing, and scientific production in this regard has increased significantly. For this reason, the small number of references used is striking, as well as the lack of depth in the information provided. I consider that it is not possible to get an accurate idea of the state of the art in this field of study.
The structure of the document does not seem coherent. The Methodology section should be placed before the Results section. This is a basic requirement that should not be omitted. It is necessary to know the design of the study before the results, in order to determine whether the results are consistent with the methodology used. The same applies to the Statistical Analysis section, which should be placed before the results. However, the Methodology section is well written and provides the necessary information for a correct understanding of the variables and measurements performed.
In relation to the sample size, I consider that the sample is too small to be able to generalize the results. Although I understand that it is not easy to access a sample in these investigations, the truth is that if we calculate the sample size for the results to be representative, it should be much larger. Therefore, I encourage the authors to continue recruiting samples in order to expand the study.
The results have been presented in a coherent manner and the figures provide clarity to the data presented.
I would recommend the authors to include sections on Limitations, Future lines of research and Practical applications. I consider that the document could be improved with this information.
Bibliography. The small number of references used is striking, considering the large amount of scientific production that has been developed in the last 5 years on this topic of study. This is perceived in the Introduction and also stands out in the Discussion, where it can be seen that an exhaustive literature review has not been carried out. This does not allow us to discuss the results adequately, providing previous and recent studies that confirm or not the results obtained.
Author Response
Answers to reviewer #1:
Thank you very much for the reviewer´s comments and detailed revision of the manuscript that indeed has help us to clarify its content and message.
Major comments
Comment 1: In the Introduction, the authors give a brief overview of this current topic. However, it is important to note that this field of study of growing interest is rapidly developing, and scientific production in this regard has increased significantly. For this reason, the small number of references used is striking, as well as the lack of depth in the information provided. I consider that it is not possible to get an accurate idea of the state of the art in this field of study.
Response: Following the reviewer’s suggestion, we have expanded the information of the state of the art regarding heart failure and gut microbiota. Thus, the Introduction section has been modified and we have included more up-to-date references.
Comment 2: The structure of the document does not seem coherent. The Methodology section should be placed before the Results section. This is a basic requirement that should not be omitted. It is necessary to know the design of the study before the results, in order to determine whether the results are consistent with the methodology used. The same applies to the Statistical Analysis section, which should be placed before the results. However, the Methodology section is well written and provides the necessary information for a correct understanding of the variables and measurements performed.
Response: We have prepared the manuscript using the template file provided by the journal, and following the remaining structure. We have included key points of our findings after the Methodology section as summary.
Comment 3: In relation to the sample size, I consider that the sample is too small to be able to generalize the results. Although I understand that it is not easy to access a sample in these investigations, the truth is that if we calculate the sample size for the results to be representative, it should be much larger. Therefore, I encourage the authors to continue recruiting samples in order to expand the study.
Response: We agree with the reviewer that our sample size is not too large, and it has been reflected in the rewritten limitations section. However, it is very similar to those of previous studies: 20 HFrEF patients/ 20 controls (ESC Heart Fail. 2017;4:282-290), 22 HF patients (PloS One. 2017;12:e0174099), 29 CHF patients/30 controls (Front Microbiol. 2022;12:813289), 30 HFpEF patients/ 30 controls (Front Cardiovasc Med. 2021;8:803744), 26 HFpEF patients/61 controls (J Am Heart Assoc. 2021;10:e020654).
The patient recruitment period was from November 2018 to January 2021. Unfortunately, the COVID-19 pandemic hindered our ability to enroll all the intended patients. Nonetheless, we believe that the longitudinal study design, combined with the cell culture experiments, can enhance the robustness of our findings, as outlined in the earlier version of the manuscript. As this remains a central focus of our research, we are presently in the process of recruiting patients with heart failure. This time, we are implementing measures to mitigate the impact of potential biases stemming from the ongoing COVID-19 pandemic.
Comment 4: I would recommend the authors to include sections on Limitations, Future lines of research and Practical applications. I consider that the document could be improved with this information.
Response: Following reviewer suggestion, we have included sections on Limitations, Future lines of research and Practical applications in the new version of the manuscript.
Comment 5: Bibliography. The small number of references used is striking, considering the large amount of scientific production that has been developed in the last 5 years on this topic of study. This is perceived in the Introduction and also stands out in the Discussion, where it can be seen that an exhaustive literature review has not been carried out. This does not allow us to discuss the results adequately, providing previous and recent studies that confirm or not the results obtained.
Response: As previously mentioned, and in response to the reviewer's request, we have expanded the information concerning the relationship between HF and gut microbiota in both the Introduction and Discussion sections. Furthermore, we have eliminated certain revisions that have been changed by the original articles.
Reviewer 2 Report
Modrego et al examined the relationship between gut microbial dysbiosis, clinical parameters and the inflammatory profile in a cohort of heart failure patients. In this longitudinal study with follow-ups at 6 and 12 months after the first diagnosis of heart failure, there was no difference in the overall gut microbiota composition but changes in the abundance of some genera were observed. The authors specifically looked for changes in gut bacteria that produce short-chain fatty acids (SCFAs) and fecal SCFAs and they showed an increase in both at the 12-month follow-up; fecal supernatant at this time point also elicited a smaller inflammatory response, as compared to that at admission, in HT29 NF-kB reporter cells.
The longitudinal design of this study provided insights into the temporal relationship between heart failure recovery and gut microbiota (the authors claimed that this was the first prospective longitudinal study of its kind). The SCFA data were largely consistent with findings from cross-sectional studies in the literature. Disappointingly, the results in the study remained largely superficial and correlative in nature. To me the authors missed an excellent opportunity to go further and explore the mechanistic link between gut microbiota and heart failure. The manuscript appeared incomplete, e.g. what components of the fecal supernatant contributed to the inflammatory response? Could SCFA or SCFA producers causally alleviate and development and/or progression of heart failure (the authors had a rat model; reference #15). Without these mechanistic data I found this manuscript is largely limited in novelty and potential in substantially advancing knowledge in this field. My other specific comments are as follows:
Major comments:
1. Please clarify the use of LEfSe to “elucidate which genera were driving divergence between admission and 12- month groups” (Lines 177-178) when there was no difference in the overall gut microbiota profile between the time points in the context of alpha and beta diversity. What was the “divergence” that the authors referred to?
2. Please clarify the “main SCFA-producing bacteria” in Line 217 and Fig 5b. What were the criteria to define “main” SCFA producers? The authors should also provide a list of the relevant bacteria.
3. Was Bifidobacterium the only genus that had significant correlation with the inflammatory response (Lines 258-259)?
4. Did the heart failure patients receive any treatments, e.g. weight management or dietary interventions, in the 12 months following the diagnosis? This might be important in explaining the change in gut microbiota.
Minor comments:
- With the small sample size I found it inappropriate to claim that the results led to the identification of “taxonomic biomarkers” (Lines 179-181).
Author Response
Answers to reviewer #2:
We greatly appreciate the reviewer's comments and the meticulous revision of the manuscript, as it has significantly contributed to enhancing the quality of the paper.
Comment 1: The longitudinal design of this study provided insights into the temporal relationship between heart failure recovery and gut microbiota (the authors claimed that this was the first prospective longitudinal study of its kind). The SCFA data were largely consistent with findings from cross-sectional studies in the literature. Disappointingly, the results in the study remained largely superficial and correlative in nature. To me the authors missed an excellent opportunity to go further and explore the mechanistic link between gut microbiota and heart failure. The manuscript appeared incomplete, e.g. what components of the fecal supernatant contributed to the inflammatory response? Could SCFA or SCFA producers causally alleviate and development and/or progression of heart failure (the authors had a rat model; reference #15). Without these mechanistic data I found this manuscript is largely limited in novelty and potential in substantially advancing knowledge in this field.
Response: As previously mentioned, and to the best of our knowledge, this study represents the first prospective longitudinal study that investigate the association between gut microbiota and alterations in bacterial metabolites in HF patients. Our findings allow us to emphasize the following key points:
- Positive clinical advancements observed within one year in new-onset HF patients are associated with the restoration of imbalanced gut microbiota.
- Certain genera of gut microbiota may function as indicators of patient progression.
- The increased production of SCFAs, notably butyrate, is linked to improvements in clinical outcomes, inflammation levels, and endothelial function in HF patients.
- In vitro experiments conducted on colonic HT29 cells exposed to fecal samples validate the reduced pro-inflammatory capacity of the gut microbiota in HF patients after one year of follow-up.
While we agree with the reviewer that establishing a mechanistic link between gut microbiota and HF requires further investigation, we plan to address this aspect in our future laboratory experiments. As previously noted, we are currently engaged in recruiting HF patients with low and high risk of readmission to examine the potential of gut microbiota as a prognostic marker and to determine the responsible components.
Regarding our HF rat model, several months ago, we initiated a new study to explore the effect of the butyrate treatment on HF development. Nevertheless, due to the extended duration required for SHHF rats to develop HF, we anticipate that these results will only be accessible next year.
We extend our gratitude to the reviewer for their invaluable assistance in propelling our research forward.
Major comments
Comment 2: Please clarify the use of LEfSe to “elucidate which genera were driving divergence between admission and 12- month groups” (Lines 177-178) when there was no difference in the overall gut microbiota profile between the time points in the context of alpha and beta diversity. What was the “divergence” that the authors referred to?
Response: Alpha diversity is a measure of microbiome inherent diversity applicable to a single community (sample), and beta diversity is a measure of the (dis)-similarity between communities (samples). These measures were computed using a rarefied OTU profile. LEfSe, an algorithm for discerning high-dimensional biomarkers and providing explanations, determines the features (organisms, taxa, OTUs, genes, functions, etc.) most likely accountable for disparities between groups. This is achieved by combining standard statistical significance tests with supplementary tests that encapsulate biological consistency and the importance of effects (Segata et al. Genome Biology 2011;12:R60). In our study, we employed LEfSe analysis to detect bacterial genera exhibiting differential abundance between fecal samples collected upon admission and those obtained after a 12-month follow-up period. The analysis was conducted using the taxonomic composition of the gut microbiota. It is worth noting that in some instances, a taxonomic group may be represented by more than one OTU. This information has been clarified in the revised manuscript and we have deleted the term divergence.
Comment 3: Please clarify the “main SCFA-producing bacteria” in Line 217 and Fig 5b. What were the criteria to define “main” SCFA producers? The authors should also provide a list of the relevant bacteria.
Response: 2. We have based the criteria on a meticulous bibliographic search of those genera with the most relevant scientific evidence. We have added a table with those that we have used in the new version of the manuscript.
Comment 4: Was Bifidobacterium the only genus that had significant correlation with the inflammatory response (Lines 258-259)?
Response: Correlation analyses were conducted using the genera identified through LEfSe analysis, namely Acidaminococcus, Bifidobacterium, Bradyrhizobium, Spingosinicella, and Sphingomonas. Among these, Bifidobacterium was the sole genus that exhibited a statistically significant correlation with inflammation. This information has been elucidated in the revised manuscript.
Comment 5: Did the heart failure patients receive any treatments, e.g. weight management or dietary interventions, in the 12 months following the diagnosis? This might be important in explaining the change in gut microbiota.
Response: Thank you very much for your comment, as it is an important clarification. Upon hospital discharge, and during the follow-up period, patients were advised to follow a Mediterranean diet, low in salt and fluids; however, no strict interventions or specific monitoring was carried out. Prior, to ensure that dietary habits of patients were not a confounding factor, participants completed the 14-item questionnaire on compliance with the Mediterranean Diet from the PREDIMED study at admission and after 12 months of follow-up. No significant changes were identified during the study period. These results have been incorporated into the revised manuscript.
Minor comments:
Comment 6: With the small sample size I found it inappropriate to claim that the results led to the identification of “taxonomic biomarkers” (Lines 179-181).
Response: We have used this term since LEfSe analysis has been described and validated as a method for metagenomic biomarker discovery (Segata et al. Genome Biology 2011;12:R60) and it is worldwide used. However, in response to the reviewer's request, we have restricted its use in the new version of the manuscript.
Round 2
Reviewer 1 Report
I consider the paper to be published. The authors have included most of the requested changes.